# Tilianin Attenuates Myocardial Ischemia–Reperfusion Injury by Targeting RIP3-Mediated Necroptosis

**DOI:** 10.3390/ph19010084

**Published:** 2025-12-31

**Authors:** Ruifang Zheng, Jie Yang, Xuemeng Wang, Yuanyuan Jin, Yue Wang, Wenling Su, Naihong Chen, Shifeng Chu, Jianguo Xing, Ming Xu

**Affiliations:** 1School of Preclinical Medicine and Clinical Pharmacy, China Pharmaceutical University, Nanjing 211198, China; zrfangdd@163.com; 2Xinjiang Key Laboratory of Uygur Medical Research, Xinjiang Institute of Materia Medica, Urumqi 830004, China; yjieer123@163.com (J.Y.); jane19573at@163.com (X.W.); wangyue20165643@163.com (Y.W.); yws990715@163.com (W.S.); chennh@imm.ac.cn (N.C.); chushifeng@imm.ac.cn (S.C.); 3Institute of Medicinal Biotechnology, Chinese Academy of Medical Sciences, Beijing 100050, China; jinyuanyuan@imb.pumc.edu.cn; 4State Key Laboratory of Bioactive Substances and Functions of Natural Medicines, Institute of Materia Medica & Neuroscience Center, Chinese Academy of Medical Sciences and Peking Union Medical College, Beijing 100050, China

**Keywords:** Tilianin, myocardial ischemic reperfusion injury, necroptosis, RIP3, CaMKII, mitochondrial membrane permeability transition pore

## Abstract

**Background/Objectives:** Necroptosis is a critical process in the pathogenesis of myocardial ischemia–reperfusion injury (MIRI). Tilianin (Til), a natural flavonoid glycoside derived from *Dracocephalum moldavica* L., exhibits significant therapeutic potential in cardiovascular diseases. However, its efficacy and mechanisms in mitigating necroptosis-induced MIRI remain incompletely understood. This study aimed to elucidate the molecular mechanisms by which Til regulates cardiomyocyte necroptosis to alleviate MIRI. **Methods:** A rat model of MIRI was established by ligating the left anterior descending coronary artery. Necroptosis in H9c2 cardiomyocytes was induced by oxygen–glucose deprivation/reoxygenation (H/R) combined with Z-VAD-FMK. Myocardial infarct size was assessed using 2,3,5-triphenyltetrazolium chloride (TTC) staining. Histopathological injury in cardiac tissue was examined by hematoxylin–eosin (HE) staining. Fluorescent probes were used to detect reactive oxygen species (ROS) and mitochondria. The molecular mechanics Poisson–Boltzmann surface area (MM-PBSA) method was used to predict the binding energy between Til and RIP3. Furthermore, RIP3 overexpression and knockdown, along with inhibition of the downstream protein CaMKII, were used to further investigate the mechanism. **Results:** Til treatment significantly reduced MIRI in rats, decreased myocardial infarct size, histopathological injury, and regulated myocardial enzyme levels. Til pretreatment effectively inhibited necroptosis in H9c2 cells induced by H/R and Z-VAD-FMK, as evidenced by reduced necroptosis rates, decreased inflammatory cytokine release, improved mitochondrial function, and suppressed phosphorylation of the necroptosis marker MLKL. Molecular docking and dynamics simulation demonstrated stable binding of Til to RIP3, which was verified through Western blot. The protective effects of Til on necroptosis were reversed by RIP3 overexpression. Furthermore, the CaMKII inhibitor KN93 abolished Til’s effect on mitochondria. **Conclusions:** Til alleviates MIRI by targeting RIP3 to inhibit the necroptosis pathway and mPTP opening. These findings provide a new therapeutic strategy for MIRI and necroptosis-related diseases.

## 1. Introduction

Thrombolysis and percutaneous coronary intervention worsen myocardial injury in patients with acute myocardial infarction, a condition known as myocardial ischemia–reperfusion injury (MIRI) [1]. However, there is no effective treatment or specific drug available. The pathogenesis of MIRI is complex, involving multiple regulated cell death pathways, excessive production of reactive oxygen species (ROS), mitochondrial dysfunction, and inflammatory responses, among others [2]. Necroptosis has been reported to determine the extent of cardiac injury caused by MIRI and is recognized as the major manner of cardiomyocyte death [3]. It plays a critical role in the development of MIRI, characterized by mitochondrial dysfunction, swelling and rupture of the cell membrane, the release of ROS, and inflammatory responses, along with increased phosphorylation of MLKL [4]. Therefore, reducing necroptosis-induced cardiomyocyte injury has become a major focus in the development of cardiovascular therapeutics.

Necroptosis accounts for approximately 70% of the myocardial injury observed during MIRI [2]. Receptor-interacting protein 3 (RIP3) is a key regulatory molecule in this process [5]. Under physiological conditions, RIP3 content in the heart is low, but it increases significantly during myocardial infarction or ischemia–reperfusion injury. Literature reports indicate that upregulation of RIP3 enhances the expression of phosphoglycerate mutase [6], leading to the opening of mitochondrial permeability transition pores (mPTPs) and triggering necroptosis [3]. In this process, calcium/calmodulin-dependent protein kinase II (CaMKII) acts as a novel substrate of RIP3. RIP3 activates the phosphorylation of CaMKII at Thr287, causing mitochondrial permeability transition pore (mPTP) opening, calcium ion (Ca^2+^) overload, and mitochondrial dysfunction, which results in energy depletion, swelling, and rupture of cardiomyocytes [7]. Ultimately, these events lead to the release of lactate dehydrogenase (LDH), decreased activity of superoxide dismutase (SOD), and cardiomyocyte necroptosis [8,9]. Thus, regulation of RIP3 activity represents a promising therapeutic strategy to reverse necroptosis during MIRI progression.

Tilianin (Til) is the principal flavonoid component of *Dracocephalum moldavica* L., exhibiting cardioprotective, anti-inflammatory, and neuroprotective properties [10]. *Dracocephalum moldavica* L. is a dried herb of the Lamiaceae family, commonly used in Uyghur and Mongolian traditional medicine for treating cardiovascular diseases (2). Various preparations of *Dracocephalum moldavica* L. have been included in the 1998 edition of the “Drug Standards of the Ministry of Health of the People’s Republic of China · Uyghur Medicine Volume,” including the single-herb preparation (Yixinbadi Ranji Buya Granules) and compound preparations (such as Aiweixin Oral Liquid) [11,12], both aimed at improving myocardial ischemia and resisting lipid peroxidation damage. As a key constituent, Til is considered a promising and safe candidate for the development of cardioprotective drugs [13]. Previous studies by this research group have demonstrated that Til exerts cardioprotective effects against MIRI through multiple mechanisms, including inhibition of mitochondrial dysfunction and inflammatory responses via modulation of CaMKII phosphorylation [14,15,16]. However, the precise mechanisms by which Til regulates the phosphorylation of CaMKII at Thr287 and its role in RIP3-mediated programmed cell death require further investigation.

In this study, it was demonstrated that Til inhibits phosphorylation of CaMKII at Thr287 and downregulates PGAM5 expression by regulating RIP3 expression. Through these mechanisms, Til inhibits mPTP opening and modulates necroptosis progression, thereby exerting protective effects against MIRI. These findings may offer a new approach to inhibit necroptosis and mitigate MIRI progression.

## 2. Results

### 2.1. Til Reduces Infarct Size and Improves Myocardial Injury in Rats with MIRI

Til significantly reduced the myocardial infarct size in Sprague-Dawley rats (Figure 1A,B). Compared with the sham group, the infarct area in the model group was markedly increased. Treatment with Til, particularly at 20 mg/kg, significantly reduced the infarct size, showing a better effect than Nec-1. HE staining analysis demonstrated that Til significantly attenuated myocardial injury (Figure 1C). At 20 mg/kg, myocardial cells exhibited more organized alignment, clearer boundaries, and a more prominent bundle-like structure. Additionally, serum biochemical analysis (Figure 1D,E) showed that LDH and CK-MB levels were significantly elevated in the model group. Treatment with Til significantly decreased these indicators, especially in the 20 mg/kg group (150.1 ± 68.8 U/L for CM-KB; 8986 ± 2311 U/L for LDH), where the reduction was similar with that observed with Nec-1 (160.3 ± 49.17 U/L for CM-KB; 11,761 ± 3516 U/L for LDH). These results collectively confirm the cardioprotective effects of Til and suggest its potential therapeutic mechanism in MIRI.

### 2.2. Til Alleviates Necroptosis and Inflammatory Response in I/R Cardiomyocyte

TUNEL staining (Figure 2A,B) revealed that the cell death rate was significantly increased in the model group. Til treatment, especially at 20 mg/kg (4.314 ± 1.465%), significantly reduced the cell death rate to a level comparable with the Nec-1 group (4.796 ± 2.431%). These findings indicate that Til may exert a protective effect on the myocardium by regulating cell death pathways. Further analysis showed that the levels of tumor necrosis factor-α (TNF-α), interleukin-1β (IL-1β), and interleukin-6 (IL-6) were significantly elevated in the serum of rats in the model group. Til administration resulted in a dose-dependent reduction in these inflammatory cytokines (Figure 2C–E). Moreover, Western blot analysis of the necroptosis marker mixed lineage kinase domain-like protein (MLKL) revealed that the level of phosphorylated MLKL (p-MLKL) was significantly increased in rats following MIRI. Pretreatment with Til markedly downregulated p-MLKL expression (Figure 2F,G). These results suggest that Til exerts anti-necroptotic and anti-inflammatory effects in myocardial ischemia–reperfusion injury.

### 2.3. Til Reduces Cell Damage and Decreases the Necrosis Rate in H9c2 Cells

Til exhibited no significant cytotoxicity toward H9c2 cells at concentrations ranging from 0.625 to 40 μg/mL (Figure 3A). In the hypoxia/reoxygenation (H/R) injury model simulating necroptosis, Til demonstrated notable cytoprotective effects (Figure 3B). Particularly, at concentrations between 2.5 and 40 μg/mL, cell viability was significantly increased. These findings suggest that Til protects H9c2 cells from necroptosis. Based on these results, Til concentrations of 2.5, 5, and 10 μg/mL were selected for subsequent experiments. Assessment of LDH release and SOD activity showed that cell injury was associated with increased LDH release and decreased SOD activity. Treatment with Til at concentrations of 2.5, 5, and 10 μg/mL significantly reduced LDH levels and enhanced SOD activity (Figure 3C,D).

To determine whether cells underwent necroptosis, Hoechst 33342/PI double staining was performed. In necroptotic cells, compromised membrane integrity allows Hoechst 33342 to stain nuclei bright blue, while propidium iodide (PI) stains necrotic cells. The results showed that Til effectively reduced the necroptosis rate of H9c2 cells at concentrations of 5–10 μg/mL (Figure 3E,F). Western blot analysis further confirmed these results, demonstrating a reduction in p-MLKL expression following Til treatment (Figure 3G,H). Additionally, detection of reactive oxygen species (ROS) using the DCFH-DA probe indicated that Til significantly decreased ROS levels, with the most pronounced effect observed at 10 μg/mL (Figure 3I). Til also showed notable anti-inflammatory effects, as indicated by reductions in inflammatory markers (Figure 3J–L).

### 2.4. Til Alleviates Mitochondrial Dysfunction in H9c2 Cells with H/R Injury

Previous studies have reported that during cardiac ischemia–reperfusion, increased glycolysis lowers intracellular pH, leading to elevated Ca^2+^ concentrations and mitochondrial Ca^2+^ overload. This triggers passive opening of mPTPs, resulting in calcium phosphate deposition, impaired ATP synthesis, and mitochondrial damage. Mitochondrial mass was assessed using the Mito Red probe, revealing that H/R injury induced mitochondrial damage, while Til treatment demonstrated protective effects (Figure 4B,D). To further investigate the mechanisms of mitochondrial damage, intracellular Ca^2+^ concentration was detected using the Fluo-4, AM probe. Til significantly reduced Fluo-4 fluorescence intensity, indicating a decrease in intracellular Ca^2+^ levels (Figure 4A,C). Subsequently, cells were labeled with the BBcellProbeTMM61 probe. In the model group, mPTP opening allowed the fluorescent quencher to enter the mitochondria, leading to a decrease in mean fluorescence intensity (MFI). Compared with the model group, Til pretreatment partially inhibited mPTP opening in H9c2 cells (Figure 4E). The effect of Til on mitochondrial membrane potential was evaluated using the JC-1 probe. Under high membrane potential, JC-1 emits red fluorescence, while under low membrane potential, it emits green fluorescence. This method effectively reflects mitochondrial functional status. In H9c2 cells subjected to H/R injury, a shift from red to green fluorescence indicated a significant decrease in mitochondrial membrane potential. However, Til pretreatment markedly reduced this fluorescence shift, suggesting an improvement in mitochondrial membrane potential (Figure 4F,G).

### 2.5. RIP 3 Is the Potential Target for Til

Til significantly enhanced the viability of H9c2 cells subjected to H/R injury, and the underlying mechanism was closely associated with the regulation of the necroptosis pathway protein RIP3 and its downstream effectors CaMKII and MLKL. Molecular docking studies revealed that Til precisely embedded into the binding pocket of RIP3 and formed hydrogen bonds with specific amino acid residues (Figure 5A,B). Further molecular dynamics simulations demonstrated that the binding between Til and RIP3 remained highly stable throughout the simulation (Figure 5C). Binding energy analysis using the MM-PBSA method indicated a strong potential inhibitory effect (Figure 5D). To validate these findings, microscale thermophoresis (MST) analysis was performed, confirming a direct interaction between Til and RIP3 with a Kd value of 5.31 nmol/L (Figure 5E). These results provide strong evidence supporting the potential development of Til as a binder of RIP3.

Western blot analysis further examined the expression levels of RIP3 and its downstream proteins. In the myocardium of rats with I/R injury, the protein levels of RIP3, (p-CaMKII and PGAM5 were significantly elevated. Pretreatment with Til markedly downregulated the expression of these proteins (Figure 5F–I). These findings were further confirmed in H9c2 cells subjected to H/R injury (Figure 5J–M), suggesting that Til may act as an inhibitor of RIP3.

### 2.6. Til Protects H9c2 Cells from H/R Injury by Inhibition of RIP3

To further verify whether Til regulates RIP3 expression, affects mitochondrial function, and thereby modulates necroptosis and inflammatory factor release, lentiviral transfection was used to overexpress RIP3 in H9c2 myocardial cells. Cells were divided into an empty vector group (Negative Control, NC) and a RIP3 overexpression group (overexpression, OE). The results showed that RIP3 overexpression led to mitochondrial damage (Figure 6A,B), which was effectively alleviated by Til treatment. ELISA analysis revealed no significant difference in LDH release and SOD activity between the RIP3 overexpression group and the model group (Figure 6C,D). However, these parameters were significantly improved following Til treatment. Further analysis of RIP3 and its downstream proteins p-CaMKII and PGAM5 (Figure 6H–K) demonstrated increased protein expression in the RIP3 overexpression group, while the protective effects of Til were diminished. Additionally, intracellular levels of inflammatory cytokines TNF-α, IL-1β, and IL-6 increased with RIP3 upregulation (Figure 6E–G). These findings indicate that Til protects mitochondria, regulates the expression of downstream pathway proteins by downregulating RIP3 expression, and ultimately reduces cell damage and inflammatory responses.

### 2.7. Inhibition of RIP 3 Protects H9c2 Cells from H/R Injury

To investigate the specific role of RIP3 in programmed necrosis of H9c2 cells induced by H/R injury, lentiviral infection was used to downregulate RIP3 expression. Cells were divided into a lentiviral empty vector group (Negative Control, NC) and a lentiviral shRNA (Knockdown, KD). Hoechst 33342/PI double staining (Figure 7A,B) showed that Til significantly reduced the necrosis rate of H9c2 cells, and RIP3 knockdown produced a similar effect. In addition, RIP3 silencing decreased LDH levels (Figure 7E), increased SOD activity (Figure 7F), protected mitochondria from damage (Figure 7C,D), and reduced the levels of inflammatory cytokines TNF-α, IL-1β, and IL-6 (Figure 7G–I). These findings provide important evidence for understanding the role of RIP3 in H/R injury.

Mechanistic studies further demonstrated that RIP3 silencing significantly reduced RIP3 protein expression. The reduction achieved by Til was not as pronounced as that induced by RIP3 silencing. Following RIP3 silencing, the expression levels of p-CaMKII and PGAM5 were also markedly downregulated (Figure 7J–M), indicating that p-CaMKII and PGAM5 are regulated downstream of RIP3 and are involved in protecting cells from programmed necrosis. Although RIP3 silencing produced a stronger effect than Til in reducing these protein levels, the mechanism by which Til protects H9c2 cells from H/R injury does not rely solely on RIP3 inhibition. Therefore, Til may exert its protective effects through additional pathways which warrant further investigation.

### 2.8. Blockade of CaMKII Abolishes the Protective Effect of Til on Cardiomyocytes Impaired by Z-VAD-FMK + H/R Injury

To verify the downstream mechanism, KN-93, a specific inhibitor of CaMKII, was employed. The experimental results showed that inhibition of CaMKII phosphorylation significantly reduced intracellular Ca^2+^ concentration (Figure 8A,C), improved mitochondrial membrane potential (Figure 8B,E) and regulated the opening of the mitochondrial permeability transition pore (Figure 8D), compared with the Z-VAD-FMK + H/R group. The attenuation on these indicators by Til was similar to that achieved with KN-93. However, no further enhancement was observed when Til was co-administered with KN-93, indicating CaMKII acts downstream mechanism of Til’s action. Western blot analysis confirmed that KN-93 effectively inhibited CaMKII phosphorylation, but no statistically significant change was observed in PGAM5. In contrast, Til significantly reduced both CaMKII phosphorylation and PGAM5 expression (Figure 8F–H). In conclusion, CaMKII regulates mitochondrial function primarily by modulating intracellular Ca^2+^ concentration and thereby influences programmed cell necrosis. However, PGAM5 regulated by Til may be independent of CaMKII.

## 3. Discussion

This study elucidates the molecular mechanism by which Til, a natural flavonoid glycoside, exerts cardioprotective effects against myocardial ischemia–reperfusion injury (MIRI) by targeting RIP3-mediated necroptosis pathway. Our results demonstrate that Til pretreatment significantly reduces myocardial infarct size, improves myocardial injury, inhibits inflammatory responses, and effectively reverses key pathological phenotypes of necroptosis in both in vivo MIRI models and H9c2 cellular H/R models. These findings collectively indicate that Til is a promising candidate for cardioprotection via specific inhibition of the necroptosis pathway.

The protective effect of Til on the myocardium post I/R is closely associated with the amelioration in core biomarkers of necroptosis. Necroptosis, a form of programmed necrosis, is characterized by loss of plasma membrane integrity, mitochondrial dysfunction, and phosphorylation/activation of MLKL [17,18]. This study confirms that Til significantly reduces the necroptosis rate in H9c2 cells induced by H/R, manifested as decreased PI uptake, directly reflecting its inhibition of membrane rupture. Concurrently, Til effectively improved mitochondrial membrane potential, suppressed mPTP opening, and reduced intracellular ROS levels. More importantly, Western Blot analysis showed that Til significantly downregulated the expression of MLKL and its phosphorylated form (p-MLKL). These results collectively indicate that Til comprehensively ameliorates multiple pathological manifestations of necroptosis, spanning from plasma membrane integrity and mitochondrial function to the terminal executioner protein.

RIP3 is a core molecule initiating the necroptotic signaling cascade [19,20]. Using multi-level experimental techniques—from computational science to biophysics and cellular signaling—this study confirms that Til is a potential direct inhibitor of RIP3. Firstly, molecular docking predicted that Til stably embeds into the binding pocket of RIP3. Subsequently, molecular dynamics simulations further confirmed the high stability of the Til-RIP3 complex throughout the 100 ns simulation, and the MM-PBSA method calculated a favorable binding free energy. Most convincingly, MST directly detected the high-affinity binding between Til and the RIP3 protein (Kd = 5.31 nM). Functionally, Til treatment significantly reduced RIP3 protein levels in both myocardial sample from MIRI rats and H/R-injured H9c2 cells. These results strongly support the hypothesis that Til acts directly on RIP3. However, the precise interaction mode and binding site require further investigation through deeper structural biology studies, such as site-directed mutagenesis or co-crystallization.

To biologically confirm that RIP3 is the key target for Til’s pharmacological action, we employed both overexpression and RNA interference knockdown strategies. The results showed that overexpressing RIP3 in H9c2 cells not only reversed Til’s protective effects on cell necrosis rate, LDH release, and SOD activity but also attenuated its inhibitory effect on downstream p-CaMKII and PGAM5 protein expression. Conversely, knocking down RIP3 using lentiviral shRNA mimicked the protective effects of Til, manifested as reduced cell necrosis, improved mitochondrial function, and decreased release of inflammatory cytokines (TNF-α, IL-1β, IL-6). These phenotypic changes observed after genetically manipulating RIP3 expression provide compelling evidence that Til’s action against cardiomyocyte necroptosis largely depends on its inhibition of RIP3.

CaMKII is a key downstream signaling molecule of RIP3 whose phosphorylation acts as a bridge mediating mitochondrial dysfunction in necroptosis [21,22]. Our study found that Til effectively inhibits RIP3 activation-induced p-CaMKII at Thr287. Experiments using the specific CaMKII inhibitor KN-93 demonstrated that inhibiting CaMKII phosphorylation could mimic Til’s effects, significantly reducing intracellular Ca^2+^ concentration, improving mitochondrial membrane potential, and regulating mPTP opening. This result confirms that inhibiting the RIP3/CaMKII signaling axis, thereby alleviating mitochondrial calcium overload and dysfunction, is one of the core mechanisms through which Til exerts its cardioprotective effects.

Although this study reveals a novel mechanism by which Til inhibits necroptosis via the RIP3/CaMKII pathway, several limitations remain. Firstly, the conclusions are primarily based on animal models and cellular experiments; the efficacy and safety of Til in humans await verification through clinical studies. Secondly, while our research focused on the RIP3-mediated non-classical necroptosis pathway, Til, as a pleiotropic natural product, might involve additional death manner, such as crosstalk with other pathological processes like autophagy or ferroptosis, which warrants future investigation.

## 4. Materials and Methods

### 4.1. Drugs and Reagents

Til (purity > 98%, determined by HPLC) was provided by the Institute of Materia Medica. Necrostatin-1 (Nec-1, T1847) was purchased from TargetMol (Shanghai, China).

### 4.2. Animals and Treatment

All animal experiments were approved by the Animal Experimental Ethics Committee of the Xinjiang Institute of Materia Medica (Approval No. XJIMM-20220102). Male Sprague-Dawley (SD) rats (6–8 weeks old) were supplied by the Experimental Animal Center of Xinjiang Medical University (License No.: SCXK (Xin) 2016-0004). Rats were housed under standard conditions (22 ± 2 °C, 45% ± 5% humidity, 12 h light–dark cycle) with free access to food and water. After acclimatization, rats were randomly divided into six groups (n = 6 per group): sham, model, Til low-dose (5 mg/kg), Til medium-dose (10 mg/kg), Til high-dose (20 mg/kg), and Nec-1 (10 mg/kg, a necroptosis inhibitor used as a positive control). Treatments were administered for seven consecutive days. Myocardial ischemia was then induced by ligating the left anterior descending coronary artery for 45 min, followed by 24 h of reperfusion [23]. Blood samples were collected from the abdominal aorta, and heart tissues were harvested, rapidly frozen in liquid nitrogen, and stored at −80 °C. All procedures complied with the National Institutes of Health guidelines for the care and use of laboratory animals.

### 4.3. Cell Culture and Viability Assay

H9c2 cells were purchased from Wuhan Procell Life Technology Co., Ltd. (Wuhan, China). and cultured in Dulbecco’s Modified Eagle Medium (DMEM, Gibco, Carlsbad, CA, USA) supplemented with 10% fetal bovine serum (FBS). Upon reaching 80–90% confluence, cells were passaged using trypsin-EDTA. To establish an in vitro H/R model simulating MIRI-related necroptosis, cells were pretreated with Z-VAD-FMK (20 μM, S7023, Selleck, Houston, TX, USA) for 24 h, followed by 6 h of oxygen–glucose deprivation and 3 h of reoxygenation. Til was administered 12 h prior to H/R induction. Cell viability was assessed using the Cell Counting Kit-8 (CCK-8, AR1160, Boster, Wuhan, China). Briefly, after treatment, cells were incubated with CCK-8 reagent for 1.5 h, and absorbance was measured at 450 nm using a microplate reader (TECAN SPARK).

### 4.4. LDH and CK-MB Determination

Blood samples were collected from the abdominal aorta and centrifuged to obtain serum, which was stored at −80 °C until analysis. The levels of lactate dehydrogenase (LDH) and creatine kinase-MB (CK-MB) were measured using commercial ELISA kits according to the manufacturers’ instructions (LDH: A020-2-2, Nanjing Jiancheng Bioengineering Institute, Nanjing, China; CK-MB: JL12296-96T, Jianglai Bio, Shanghai, China). Moreover, the leakage of LDH from H9c2 cells was determined as LDH leakage ratio, calculated as LDH in the medium/(LDH in the medium + LDH in cell lysate) × 100%

### 4.5. Assessment of Myocardial Infarct Size

Myocardial infarct size was evaluated by 1% TTC staining. Briefly, heart tissues were sectioned into 2 mm thick slices and incubated with TTC solution for 10–15 min at 37 °C. Viable myocardium stained red, while infarcted areas remained pale. Stained sections were scanned, and the infarct area was quantified using ImageJ software. The entire left ventricular area was considered the region at risk due to the LAD ligation model. The infarct area ratio was calculated as (infarct area/total left ventricular area) × 100%.

### 4.6. Histopathological Examination

Hearts were harvested after reperfusion, fixed in 10% neutral buffered formalin, and embedded in paraffin. Sections (5 μm thickness) were stained with HE and observed under a light microscope (LEICA DM 3000 LED, Wetzlar, Germany) at 20× magnification to assess myocardial injury.

### 4.7. Detection of Cardiomyocyte Death

Cardiomyocyte death was detected using a TUNEL assay kit (C1086, Beyotime, Shanghai, China) according to the manufacturer’s protocol. Briefly, paraffin-embedded heart sections were deparaffinized, stained with TUNEL reagent at 37 °C, and counterstained with DAPI (E-CK-A321, Elabscience, Wuhan, China). Images were captured using a fluorescence microscope (Evos FLoid, Thermo Fisher, MA, USA), and TUNEL-positive cells were quantified.

### 4.8. Measurement of Mitochondrial Membrane Potential

Mitochondrial membrane potential was assessed using the JC-1 assay (C2006, Beyotime, Shanghai, China). H9c2 cells were seeded in 6-well plates and incubated with JC-1 staining solution at 37 °C for 20–30 min. After washing with JC-1 buffer, cells were observed under a fluorescence microscope. The ratio of red (aggregates) to green (monomers) fluorescence intensity was calculated to evaluate mitochondrial depolarization.

### 4.9. Hoechst 33342/PI Staining for Necroptosis

H9c2 cells were seeded in 6-well plates at a density of 1 × 10^5^ cells per well. After 24 h, cells were washed with PBS and stained with Hoechst 33342 (1:1000, 875756-97-1, Sigma, MO, USA) and propidium iodide (1 mg/mL, PI, 2269742-45-7, Sigma, MO, USA) for 20–30 min at 37 °C. Cells were imaged using a fluorescence microscope, and necroptotic cells (positive for both bright blue Hoechst and red PI staining) were quantified with ImageJ software.

### 4.10. Measurement of mPTP Opening

Mitochondria permeability transition pore (mPTP) opening was assessed using BBcellProbe M61 kit (BestBio, Shanghai, China) as reported previously. The fluorescence intensity of BBcellProbe M61 probe was measured on a BioTek Epoch microplate reader (TECAN SPARK, Männedorf, Switzerland).

### 4.11. Determination of Intracellular Ca^2+^

H9c2 Cell pellets following two PBS washes were incubated with 4 μΜ Fluo-4 AM for 20 min at 37 °C under protection from light. Following 3 washes with chilled HEPES buffer and filtering with a 300-mesh sieve, the fluorescence intensity was analyzed using a BD FACSCalibur flow cytometer.

### 4.12. Molecular Docking and Dynamics Simulation

The crystal structure of RIP3 (PDB ID: 7MX3) was obtained from the Protein Data Bank. All non-essential molecules were removed. Tilianin was prepared and docked into the RIP3 binding site using MGLTools 1.5.6 and AutoDock Vina 1.2.3. The conformation with the highest predicted binding affinity was selected for further analysis.

Molecular dynamics (MD) simulations were performed with GROMACS 2019.6 using the AMBER14SB and GAFF force fields. The protein–ligand complex was solvated in a TIP3P water box, neutralized with counter ions, and energy-minimized. After NVT and NPT equilibration, a 100 ns production MD simulation was conducted. The root-mean-square deviation (RMSD) of protein backbone atoms was calculated, and the binding free energy between Til and RIP3 was estimated using the MM-PBSA method. Additionally, the interaction was validated by MST.

### 4.13. Overexpression and Knockdown of RIP3

To investigate the role of RIP3, H9c2 cells were infected with RIP3 overexpression or silencing lentiviruses (GeneChem) according to the manufacturer’s instructions. Transfected cells were used to evaluate the effects of RIP3 modulation on CaMKII phosphorylation, mitochondrial function, and inflammatory cytokine release.

### 4.14. Blockade of CaMKII

To investigate the role of phosphorylation of CaMKII in the protection of Til on cardiomyocytes impaired by Z-VAD-FMK + H/R injury, H9c2 cells were cultured with KN93 (HY-15465, MCE, Shanghai, China). These cells were used to evaluate the effects of phosphorylation of CaMKII on intracellular Ca^2+^, mitochondrial function, and the expression of PGAM5.

### 4.15. Western Blot Analysis

Total protein was extracted from cells using RIPA lysis buffer containing protease and phosphatase inhibitors (P6731, Solarbio, Beijing, China). Protein concentration was determined with a BCA assay kit (A55865, Thermo Fisher, Waltham, MA, USA). Proteins were separated by SDS-PAGE and transferred to PVDF membranes. After blocking with 5% non-fat milk, membranes were incubated overnight at 4 °C with primary antibodies against RIP3 (1:1000, ab62344, Abcam, Waltham, MA, USA), CaMKII (1:1000, ab134041, Abcam, MA, USA), p-CaMKII (1:1000, ab182647, Abcam, MA, USA), MLKL (1:1000, ab243142, Abcam, MA, USA), p-MLKL (1:1000, 26539, Cell Signaling Technology, Danvers, MA, USA), and PGAM5 (1:1000, ab3222067, Abcam, MA, USA). β-Tubulin (1:5000, AC012, Abclone, Wuhan, China) was used as a loading control. Membranes were then incubated with HRP-conjugated secondary antibodies (1:2000, ZB-2306, Zhongshan Golden Bridge, Beijing, China), and protein bands were visualized using a chemiluminescence detection system. Band intensities were quantified with ImageJ 1.54g software and normalized to tubulin.

### 4.16. Statistical Analysis

Data are presented as the mean ± standard deviation (SD). Statistical analyses were performed using GraphPad Prism 8.0. Differences between two groups were analyzed by two-tailed Student’s *t*-test for normally distributed data. Multiple group comparisons were conducted using one-way analysis of variance (ANOVA) followed by an appropriate post hoc test. *p* < 0.05 was considered statistically significant.

## 5. Conclusions

In summary, this work demonstrates that Til alleviates MIRI by directly targeting RIP3, inhibiting its kinase activity and subsequent CaMKII phosphorylation, thereby improving mitochondrial function, reducing ROS release, and ultimately blocking the execution of cardiomyocyte necroptosis. This mechanism not only provides a novel theoretical basis for understanding Til’s cardioprotective effects, but also lays a solid experimental foundation for its potential development as a therapeutic agent for MIRI and other necroptosis-related diseases.

## Figures and Tables

**Figure 1 pharmaceuticals-19-00084-f001:**
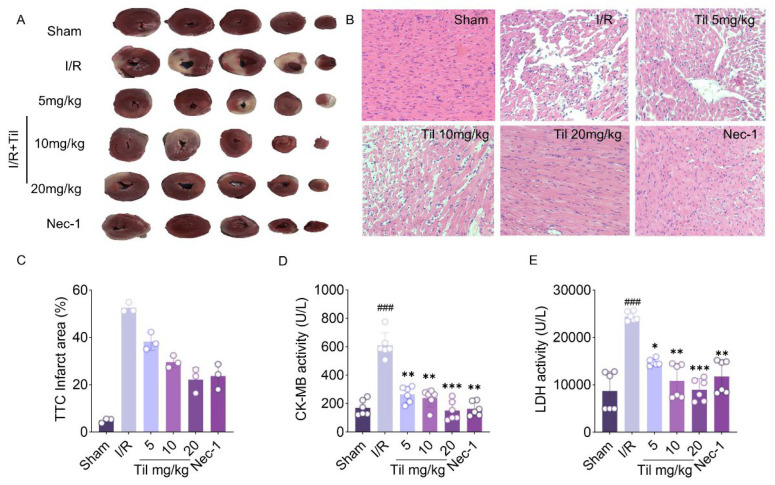
Til reduces infarct size and improves myocardial injury in rats with MIRI. (**A**) Representative images of cardiac cross-sections stained with TTC; (**B**) Histopathological staining of myocardial tissues (20×); (**C**) Percentage of infarct area determined by TTC-staining (n = 3); (**D**,**E**) Serum levels of CK-MB (**D**) and LDH (**E**) in each group measured by ELISA (n = 6). ^###^
*p* < 0.001 vs. sham group; * *p* < 0.05, ** *p* < 0.01, *** *p* < 0.001 vs. I/R group.

**Figure 2 pharmaceuticals-19-00084-f002:**
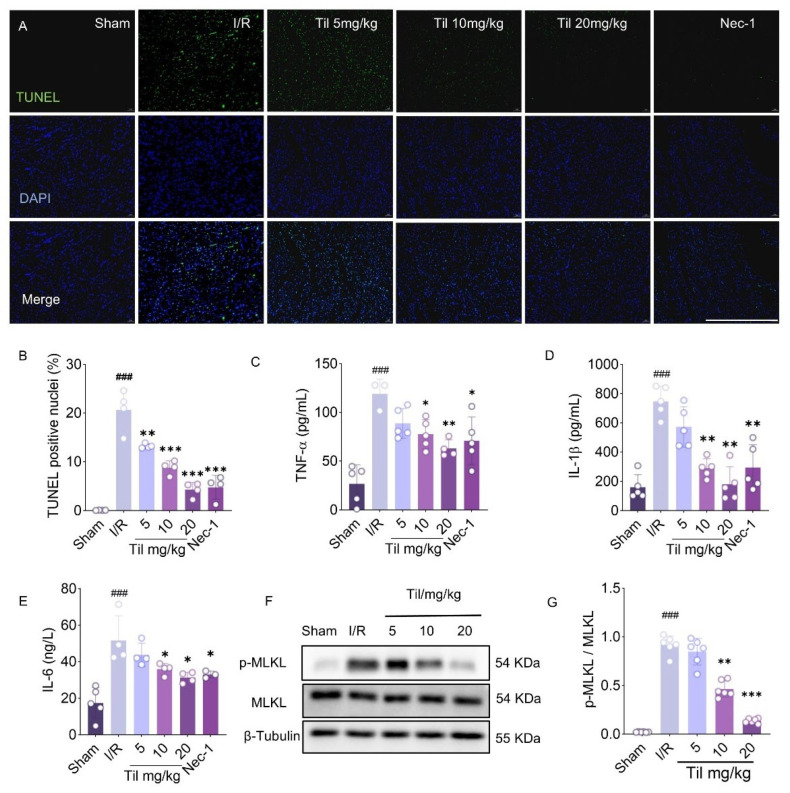
Til alleviates necroptosis and inflammatory response in rat with MIRI. (**A**,**B**) TUNEL staining (magnification ×20) of myocardial tissues and positive staining rate (n = 4; Scale bar: 500 μm); (**C**–**E**) Serum levels of TNF-α, IL-1β, and IL-6 detected by ELISA (n = 5); (**F**) Representative blots of MLKL and p-MLKL; (**G**) Quantitative analysis of Western blot intensity using ImageJ (n = 6). ^###^
*p* < 0.001 vs. control group; * *p* < 0.05, ** *p* < 0.01, *** *p* < 0.001 vs. model group.

**Figure 3 pharmaceuticals-19-00084-f003:**
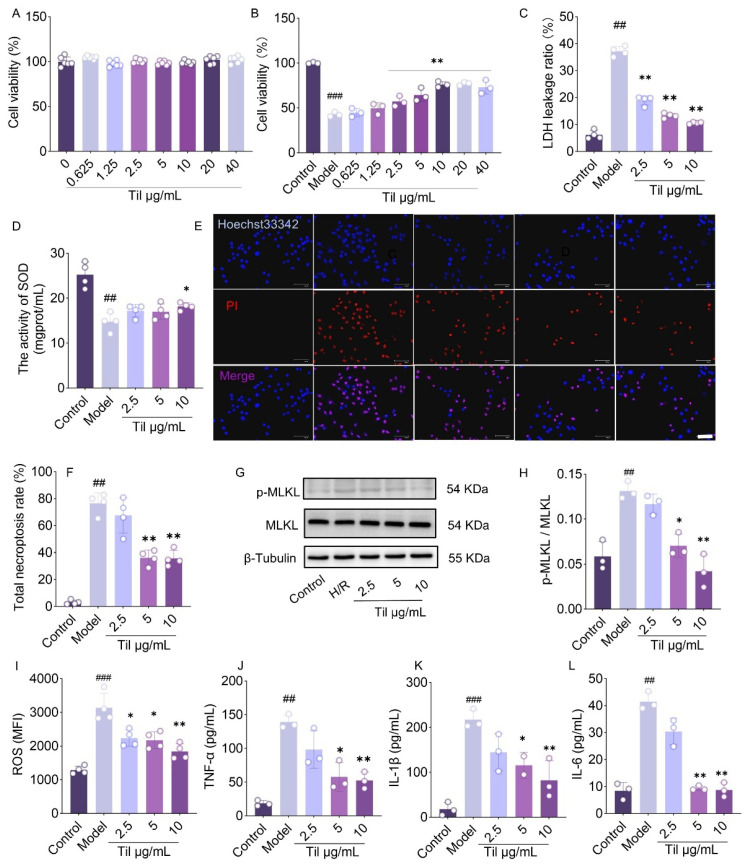
Til reduces myocardial cell damage and decreases the necrosis rate in vitro. (**A**,**B**) Effects of Til at different concentrations (0.625–40 μg/mL) on the viability of H9c2 cells following hypoxia/reoxygenation (H/R, n = 6 in (**A**) and n = 3 in (**B**)); (**C**,**D**) Measurement of LDH leakage ratio and SOD levels (n = 4); (**E**,**F**) Detection of the necroptosis rate in H9c2 cells using Hoechst 33342/PI double staining (scale bar = 100 μm, n = 3); (**G**,**H**) Western blot bands of MLKL and phosphorylated MLKL (p-MLKL) and corresponding quantitative analysis (n = 3); (**I**) Detection of ROS in H9c2 cells using the DCFH-DA probe (n = 4); (**J**–**L**) Measurement of TNF-α, IL-1β, and IL-6 levels using ELISA (n = 3). ^##^
*p* < 0.01, ^###^
*p* < 0.001 vs. control group; * *p* < 0.05, ** *p* < 0.01, vs. model group.

**Figure 4 pharmaceuticals-19-00084-f004:**
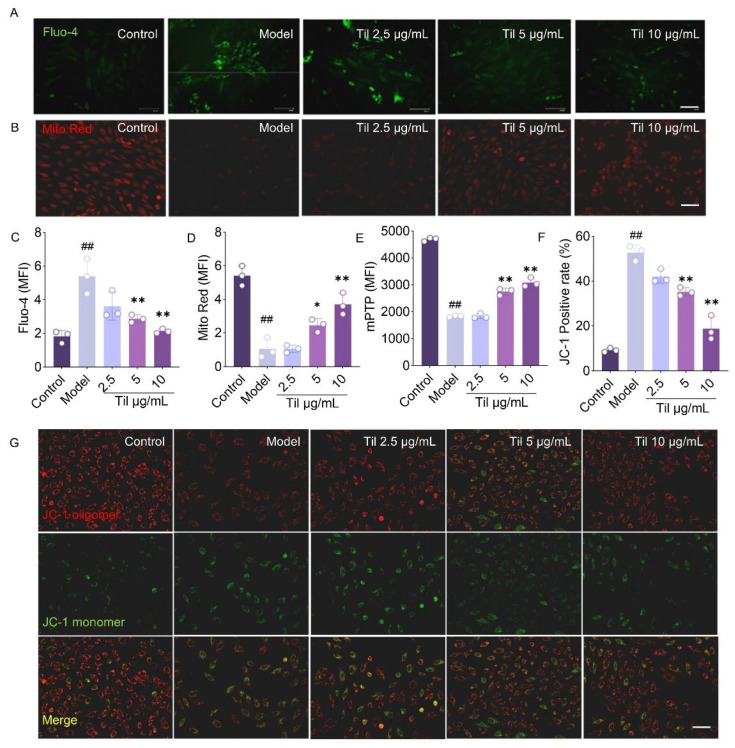
Til alleviates mitochondrial dysfunction in H9c2 cells following H/R injury. (**A**,**C**) Measurement of intracellular Ca^2+^ concentration in H9c2 cells using the Fluo-4-AM probe (n = 3; Scale bar: 30 μm); (**B**,**D**) Detection of mitochondria using Mito Tracker Red (n = 3; Scale bar: 30 μm); (**E**) Labeling and detection of mitochondrial permeability transition pore opening using the BBcellProbeMM61 probe; (**F**,**G**) Labeling of mitochondrial membrane potential using JC-1 (n = 3; Scale bar: 30 μm). ^##^
*p* < 0.01 vs. control group; * *p* < 0.05, ** *p* < 0.01 vs. model group.

**Figure 5 pharmaceuticals-19-00084-f005:**
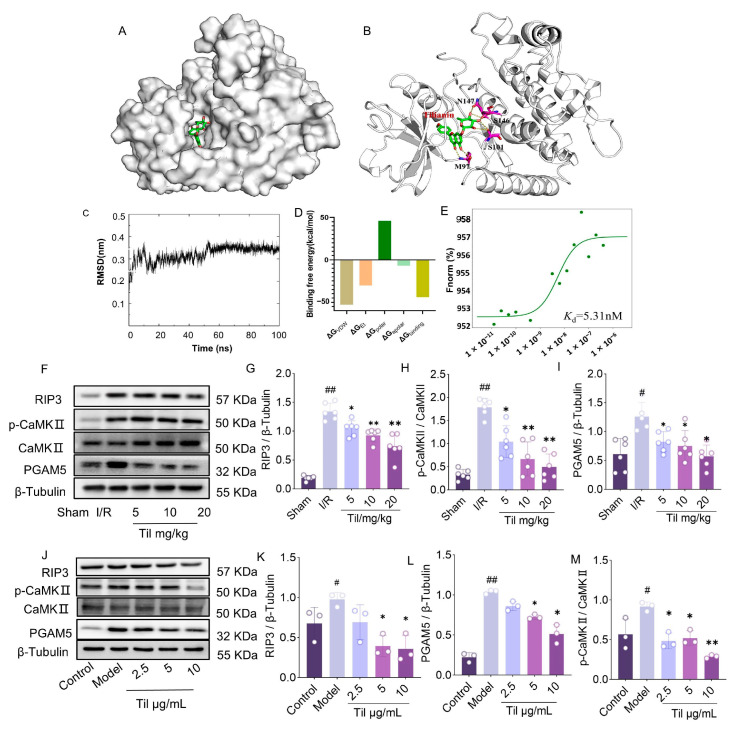
RIP3 is potential target of Til. (**A**) Schematic diagram of molecular docking between Til and RIP3.The left picture is the whole view, and the right picture is the partial view. The green stick is Til, the cyan cartoon is protein, the purple stick is the amino acid that Til forms hydrogen bonds with RIP3, and the dotted line indicates hydrogen bonding. (**B**) Molecular docking shows that Til forms hydrogen bonds with RIP3; (**C**) RMSD curve of Til and RIP3 complex; (**D**) Calculation of binding free energy; (**E**) Confirmation that RIP3 is the direct target of Til using MST (n = 3); (**F**) Western blot detection of RIP3, phosphorylated CaMKII (p-CaMKII), and PGAM5 protein expression in rat myocardial tissue; (**G**–**I**) Quantitative analysis of corresponding protein expression in rat myocardium using ImageJ (n = 6); (**J**) Western blot detection of RIP3, p-CaMKII, and PGAM5 protein expression in H9c2 cells; (**K**–**M**) Quantitative analysis of corresponding protein expression in H9c2 cells using ImageJ 1.54g (n = 3). ^#^ *p* < 0.05, ^##^ *p* < 0.01 vs. sham or control group; * *p* < 0.05, ** *p* < 0.01 vs. I.R or model group.

**Figure 6 pharmaceuticals-19-00084-f006:**
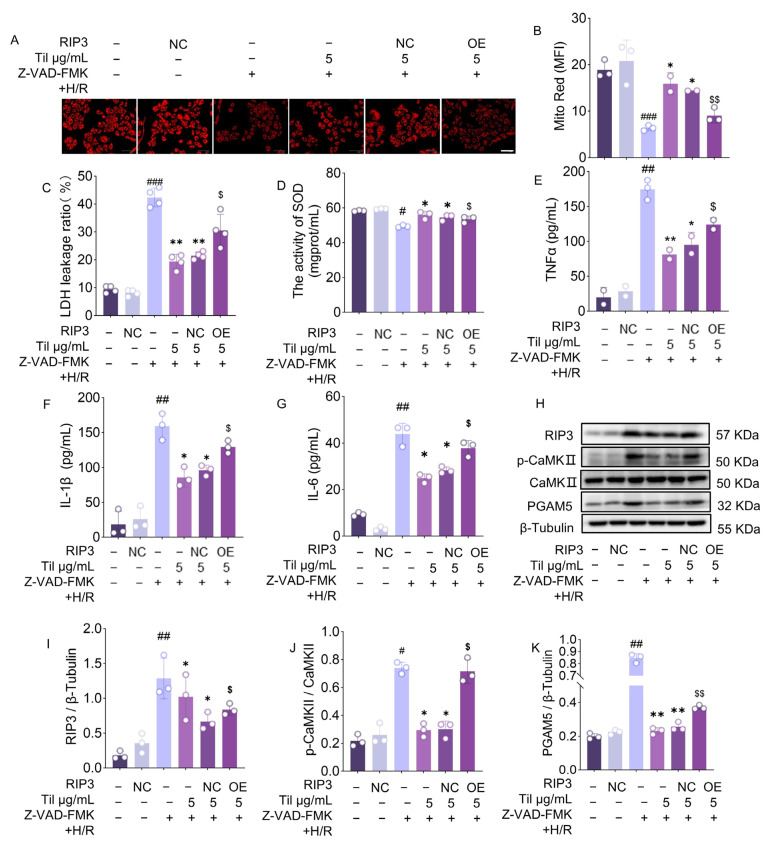
Til protects myocardial cells from H/R injury by inhibiting RIP3 expression. (**A**,**B**) Effects of RIP3 on mitochondria in H9c2 cells (n = 3; Scale bar: 30 μm); (**C**,**D**) Measurement of LDH leakage ratio (n = 4) and SOD levels (n = 3); (**E**–**G**) Detection of TNF-α, IL-1β, and IL-6 release by ELISA (n = 3); (**H**) Western blot analysis of RIP3, phosphorylated CaMKII (p-CaMKII), PGAM5, and phosphorylated MLKL (p-MLKL) in H9c2 cells after RIP3 overexpression; (**I**–**K**) Quantitative analysis of the corresponding Western blot bands using ImageJ (n = 3). ^#^ *p* < 0.05, ^##^ *p* < 0.01, ^###^ *p* < 0.001 vs. control group; * *p* < 0.05, ** *p* < 0.01 vs. model group; ^$^ *p* < 0.05, ^$$^ *p* < 0.01 vs. Til group.

**Figure 7 pharmaceuticals-19-00084-f007:**
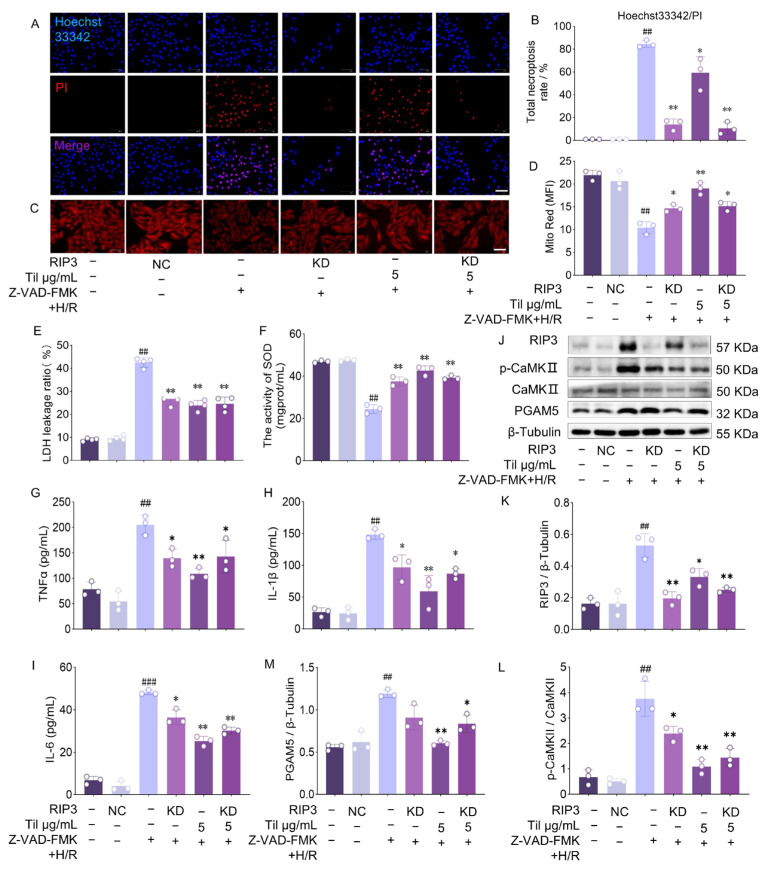
Inhibition of RIP3 protects cardiomyocytes from H/R injury. (**A**,**B**) Effect of RIP3 on the rate of programmed necrosis (n = 3; Scale bar: 100 μm); (**C**,**D**) Effect of RIP3 on mitochondria in H9c2 cells (n = 3; Scale bar: 30 μm); (**E**,**F**) Measurement of LDH leakage ratio (n = 4) and SOD levels (n = 3); (**G**–**I**) Detection of TNF-α, IL-1β, and IL-6 release by ELISA (n = 3); (**J**) Western blot analysis of RIP3, phosphorylated CaMKII (p-CaMKII), PGAM5, and phosphorylated MLKL (p-MLKL) in H9c2 cells after RIP3 interference; (**L**,**M**) Quantitative analysis of Western blot bands of the ratio of p-CaMKII/CaMKII (**L**), RIP3 (**K**) and PGAM5 (**M**) using ImageJ 1.54g (n = 3). ^##^ *p* < 0.01, ^###^
*p* < 0.001 vs. control group; * *p* < 0.05, ** *p* < 0.01 vs. Z-VAD-FMK + H/R group.

**Figure 8 pharmaceuticals-19-00084-f008:**
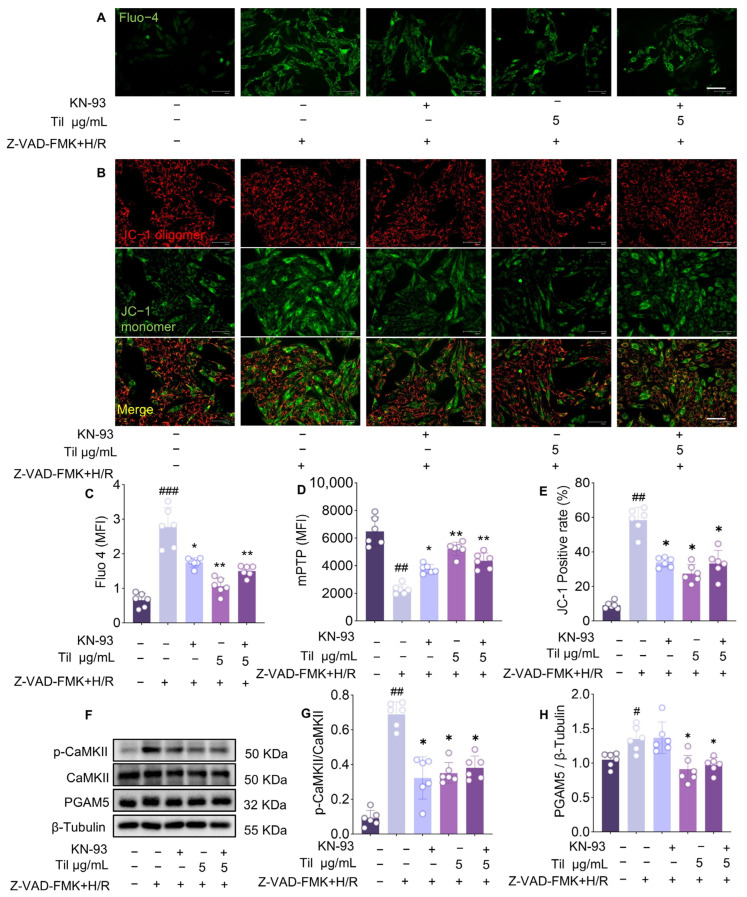
Blockade of CaMKII abolished the protection of Til on cardiomyocytes impaired by Z-VAD-FMK + H/R injury. (**A**,**C**) The representative images of intracellular Ca^2+^ concentration in H9c2 cells was measured using the Fluo-4-AM probe and their qualitive analysis (n = 6; Scale bar: 30 μm); (**B**,**E**) The representative images of mitochondrial membrane potential was labeled with JC-1 and their qualitive analysis (n = 6; Scale bar: 30 μm); (**D**) Labeling and detection of mitochondrial permeability transition pore opening using the BBcellProbeMM61 probe (n = 6); (**F**) Representative immunoblots of p-CaMKII and PGAM5 in H9c2 cells; (**G**,**H**) Quantitative data of the corresponding Western blot intensity measured by Image J (n = 6). ^#^
*p* < 0.05, ^##^
*p* < 0.01, ^###^
*p* < 0.001 vs. control group; * *p* < 0.05, ** *p* < 0.01 vs. model group.

## Data Availability

The original contributions presented in this study are included in the article. Further inquiries can be directed to the corresponding authors.

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
