# Peer review of "Tilianin Attenuates Myocardial Ischemia–Reperfusion Injury by Targeting RIP3-Mediated Necroptosis"

_pharmaceuticals, 2025, doi:10.3390/ph19010084_

Round 1
Reviewer 1 Report
Comments and Suggestions for Authors
The manuscript of the authors is devoted to an interesting and relevant research topic. The article is well written, in good language. The article represents a serious study, the methods adequately meet the goals and objectives. The topic investigated in this manuscript is highly relevant to current cardiovascular research. The role of necroptosis in myocardial ischemia-reperfusion injury has gained substantial recognition in recent years, and the exploration of natural compounds like Tilianin as potential inhibitors of this cell death pathway represents an important contribution to the field of cardioprotection.
However, it has a number of shortcomings that require correction and clarification.
Major questions
- On what basis were these specific doses of tilianin chosen (5, 10, 20 mg/kg in vivo and 2.5-10 μg/mL in vitro)? Were preliminary studies conducted to determine the optimal doses? Are these experimental doses aligned with those used in clinical practice or traditional medicine? Furthermore, it remains unclear whether a broader dose-range screening was performed to identify optimal efficacy and potential toxicity thresholds. Additional emphasis on dose-response relationships in the results section would strengthen the pharmacological relevance of the findings.
- Why was the combination of H/R + Z-VAD-FMK used to induce necroptosis? What evidence confirms that this model specifically induces necroptosis and not other forms of cell death?
- Were experiments performed with inhibitors of other forms of cell death (apoptosis, autophagy) to confirm the specificity of tilianin's action on necroptosis?
- Insufficient experimental data are presented to definitively conclude that mPTP opening is specifically inhibited. While the data from the BBcellProbe™61 fluorescent probe suggest Tilianin modulates mitochondrial permeability, definitive conclusions regarding specific mPTP inhibition require validation through complementary direct approaches. Recommended methodological enhancements include:
- Quantitative assessment of mitochondrial swelling via spectrophotometry
- Monitoring cytochrome c release as a downstream indicator of mPTP opening
- Pharmacological verification using established mPTP inhibitors (e.g., cyclosporine A) to confirm mechanism specificity
Minor/technical questions
- The Materials and Methods section, in my view, requires improvement. The methodological descriptions lack completeness, as specific doses and concentrations of drugs and reagents are omitted. Specifically, for the Western blotting analysis, it is essential to provide:the commercial sources of all primary and secondary antibodies, their respective catalog numbers, and the dilutions employed. In particular, it is unclear which form of tubulin antibodies was used for normalization (α,β?). This information must be specified.
- Figure 1c It is necessary to specify what the percentages are calculated relative to. It is unclear what the images in Figure 1A represent – whether they show a decrease over time, or illustrate the range of values. Please clarify.
Reviewer 2 Report
Comments and Suggestions for Authors
This is an interesting paper investigating the effects and mechanisms of action of TIL on MIRI in vivo and H/R injury in vitro. I have a few major and several minor comments or requirements:
Major comments:
Result section:
- n=3 is not sufficient for a statistical analysis: ANOVA requires normal distribution and equal variances. The results in Fig. 8 are partially questionable in this respect. The data points in TIL vrs. TIL+KN-93 are really close together in some figures so that with n=3 one would not ground a serious conclusion based on this experiment with such a low sample number. Moreover, the semi-quantitative analysis of western blots is subject to inaccuracies. In this respect, the person doing the densitometric analysis should be blinded (not knowing which signal is from which group). I require increasing the number of samples at least to n=6 for Fig. 8 and a blinded evaluation.
- Further, the number of stars in many of the other figures with n= 3 visually overestimate the results with respect to likelihoods/probability. I would suggest again either increasing the sample number or doing without any statistics at all when sample number is no more than n=3. In many of your figures one can already see the effect clearly (all 3 data points of one group very high and all 3 of another very low). The statistics would not strengthen the results at all any further. I would suggest not using so many stars in this instance.
- To investigate necrosis in cell culture by measuring LDH, it is necessary to normalize LDH levels in the medium to the LDH-content (LDH-ratio), since the number of cells and their metabolic state must be considered (more or less release into the medium despite same degree of damage).
Minor comments/questions
Methods:
- To compare infarct sizes, it is usually necessary to normalize the infarcted area to the “area of risk” (the perfusion field of the coronary artery which becomes ligated), thereby considering that there are individual differences in coronary perfusion and thus infarct sizes may be different already due to this fact. Why was this aspect neglected?
- In figures 1 and 2 Nec-1 was used as “positive control”. Why did the authors not follow on with Nec-1 in the experiments in Figure 3 ff?
Results
- Please insert a scale-bar (length) within each picture.
- Text lines 104-106 and Fig. 2: The conclusion (greater reduction by TIL vrs. Nexc-1) cannot be made. A separate statistical evaluation (-fold differences to control / model) would be needed and, moreover, it doesn’t really look like there was a difference.
- Fig 3C and 6C: LDH-release needs to be normalized to LDH content (or to total LDH levels)
References
In all references within the titles and at the end strange numbers appear (from 480 top 501) in the file I got. I believe this was not intended?
Round 2
Reviewer 2 Report
Comments and Suggestions for Authors
The authors have successfully answered all my concerns. Nice work!